# Reproducing DragDiffusion: Interactive Point-Based Editing with Diffusion Models

## Abstract

DragDiffusion (Shi et al., 2024) is a diffusion-based method for interactive point-based image editing that enables users to manipulate images by directly dragging selected points. The method claims that accurate spatial control can be achieved by optimizing a single diffusion latent at an intermediate timestep, together with identity-preserving fine-tuning and spatial regularization. This work presents a reproducibility study of DragDiffusion using the authors' released implementation and the DragBench benchmark. We reproduce the main ablation studies on diffusion timestep selection, LoRA-based fine-tuning (Hu et al., 2022), mask regularization strength, and UNet feature supervision, and observe close agreement with the qualitative and quantitative trends reported in the original work. At the same time, our experiments show that performance is sensitive to a small number of hyperparameter assumptions, particularly the optimized timestep and the feature level used for motion supervision, while other components admit broader operating ranges. We further evaluate a multi-timestep latent optimization variant and find that it does not improve spatial accuracy while substantially increasing computational cost. Overall, our findings support the central claims of DragDiffusion while clarifying the conditions under which they are reliably reproducible.

## 1 Introduction

Diffusion models have become a dominant paradigm for high-quality image generation and editing, supporting tasks such as inpainting, image-to-image translation, and text-guided manipulation. While these models excel at capturing rich visual distributions, they offer limited mechanisms for specifying precise geometric intent. Text-based prompts provide flexible semantic control but are often insufficient for enforcing exact spatial transformations. Interactive editing interfaces that allow users to directly manipulate image structure therefore represent an important complementary direction.

DragDiffusion (Shi et al., 2024) proposes an interactive, point-based editing framework built on diffusion models, in which users specify desired motion by dragging handle points to target locations. Rather than optimizing along the full diffusion trajectory, the method performs latent optimization at a single, carefully chosen diffusion timestep, using motion supervision on UNet feature activations to guide spatial changes. Identity-preserving fine-tuning via low-rank adaptation and spatial mask regularization are incorporated to stabilize edits and prevent unintended global distortions.

The effectiveness of DragDiffusion depends on several tightly coupled design choices, including the optimized diffusion timestep, the strength and duration of identity-preserving fine-tuning, the weight of spatial regularization, and the UNet feature level used for motion supervision. From a reproducibility perspective, this combination of diffusion inversion, latent-space optimization, and feature-level supervision introduces multiple sources of sensitivity. In particular, optimization stability and performance can be strongly affected by timestep selection and scheduler behavior, and small configuration changes may lead to divergent outcomes. Although the authors release both an implementation and the DragBench benchmark for evaluation, the robustness of the reported empirical behavior under such sensitivities warrants systematic examination.

In this work, we present a reproducibility study of DragDiffusion aimed at validating its empirical claims and clarifying the sensitivity of its core design decisions. Using the authors' released codebase and evaluation protocol, we replicate the main ablation studies reported in the original paper, covering diffusion timestep selection, LoRA-based fine-tuning (Hu et al., 2022), mask regularization strength, and UNet feature supervision. We further introduce a controlled extension that jointly optimizes multiple diffusion timesteps to assess whether the single-timestep strategy is essential or merely a convenient approximation.

Our results reproduce the qualitative and quantitative trends reported in the original work and show that intermediate-timestep optimization combined with identity-preserving fine-tuning and mid-level feature supervision yields the most reliable spatial control. At the same time, extending optimization across multiple timesteps increases computational cost without improving performance. Together, these findings support the original design choices while highlighting practical considerations that are critical for reliable reproduction and future work on interactive diffusion-based image editing.

To contextualize the scope of this reproducibility study, the original DragDiffusion paper reports a series of quantitative ablation experiments evaluating **(i)** the choice of diffusion timestep for latent optimization, **(ii)** the presence and extent of identity-preserving fine-tuning via LoRA, **(iii)** the strength of spatial mask regularization, and **(iv)** the UNet feature level used for motion supervision, alongside extensive qualitative comparisons and benchmark evaluations on DragBench. In this work, we focus on reproducing all core quantitative ablations reported in the original paper under the same evaluation protocol. We do not reproduce large-scale qualitative comparisons or user studies, as these are inherently subjective and less amenable to exact replication. In addition to reproducing the original ablations, we introduce a controlled extension that jointly optimizes multiple diffusion timesteps in order to test the necessity of the single-timestep design choice proposed by the authors.

For clarity and ease of comparison, each reproduced experiment in this paper is explicitly cross-referenced to the corresponding figure or table in the original DragDiffusion paper (Shi et al., 2024). This allows readers to directly align our reproduced results with the original findings without reproducing copyrighted figures or tables.

## 2 Scope of Reproducibility

This reproducibility study is structured around a targeted evaluation of the empirical evidence supporting DragDiffusion (Shi et al., 2024). Rather than re-deriving the method or proposing architectural modifications, our objective is to assess whether the experimental results reported in the original paper can be independently recovered under the same evaluation protocol and to determine how sensitive those results are to key implementation and hyperparameter choices.

Concretely, we reproduce the principal experimental claims articulated in the original work by isolating and evaluating each corresponding design component under controlled conditions. The claims we examine are:

- **Claim 1:** The choice of diffusion timestep at which latent optimization is performed has a decisive impact on spatial accuracy and stability, with intermediate timesteps yielding the best performance.

- **Claim 2:** Identity-preserving fine-tuning via low-rank adaptation is required to prevent identity drift and degradation of visual quality during point-based edits.

- **Claim 3:** Editing performance varies with the extent of identity-preserving fine-tuning, exhibiting diminishing improvements beyond a moderate number of optimization steps.

- **Claim 4:** Spatial mask regularization plays a critical role in constraining edits and mediating the trade-off between localized control and global image fidelity.

- **Claim 5:** The UNet feature level used for motion supervision substantially affects spatial precision, with mid-level decoder features providing the most effective guidance.

Beyond direct replication of these claims, we include an auxiliary experiment that modifies the optimization procedure by jointly updating multiple diffusion latents within a single editing run. This extension is designed to probe whether the reported single-timestep strategy is empirically sufficient or whether additional supervision across timesteps can yield measurable benefits under the same evaluation setting.

## 3 Methodology

This section describes the methodology used to reproduce and evaluate the central claims of DragDiffusion (Shi et al., 2024). Our objective is to faithfully execute the authors' released implementation under controlled experimental settings, while systematically analyzing the impact of the key design choices highlighted in the original paper.

Figure 1 illustrates a high-level overview of the DragDiffusion pipeline as reproduced in this study.

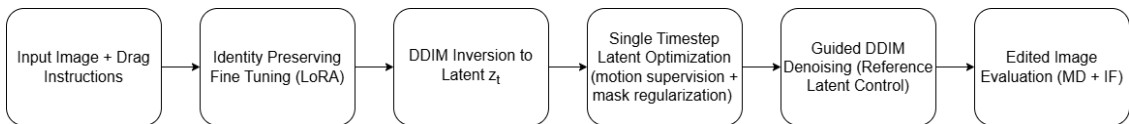

Figure 1: Overview of the DragDiffusion pipeline reproduced in this study. Given an input image, prompt, mask, and handle–target point pairs, the image is inverted using DDIM. A single latent at timestep $t$ is optimized via motion supervision and mask regularization. The optimized latent is then used during forward DDIM denoising with attention control to generate the edited image. Identity-preserving LoRA weights are optionally applied during generation.

### 3.1 Overview of DragDiffusion

DragDiffusion enables interactive image editing by allowing users to specify pairs of handle and target points that encode desired spatial motion. Given an input image and a text prompt, the method first performs deterministic DDIM inversion (Song et al., 2021) to obtain a latent representation at a selected diffusion timestep.

Latent optimization is then performed to enforce semantic correspondence between handle points and their target locations using motion supervision applied to UNet feature maps. The optimized latent is subsequently used as a reference during guided DDIM denoising to generate the final edited image.

All experiments in this work are conducted using Stable Diffusion v1.5 (Rombach et al., 2022), which serves as the underlying diffusion backbone.

### 3.2 Identity-Preserving Fine-Tuning

Following (Shi et al., 2024), identity preservation is achieved via Low-Rank Adaptation (LoRA) (Hu et al., 2022) applied to the attention layers of the UNet architecture (Ronneberger et al., 2015). LoRA enables parameter-efficient fine-tuning by introducing low-rank updates to the attention weights while keeping the original model parameters frozen.

We implement LoRA fine-tuning using the authors' released training routine with the following configuration:

- LoRA rank fixed to 16
- Learning rate set to $5 \times 10^{-4}$
- Fine-tuning performed for up to 120 optimization steps

To improve computational efficiency, we reuse existing LoRA checkpoints whenever available and only retrain samples missing the required checkpoints. This modification does not alter the optimization objective and preserves the intended identity-preserving behavior.

### 3.3 Single-Timestep Latent Optimization

A core design principle of DragDiffusion is to optimize the latent representation at a single diffusion timestep $t$. After DDIM inversion, the latent corresponding to timestep $t$ is iteratively updated by minimizing a motion supervision loss defined over UNet feature activations.

Specifically, semantic features extracted at handle point locations are encouraged to match features at the corresponding target locations. A spatial mask regularization term constrains updates to a predefined editable region, preventing unintended global distortions.

Unless otherwise stated, we follow the authors' recommendation and fix the optimized timestep to $t = 35$, which corresponds to an intermediate noise level in the diffusion process.

### 3.4 Ablation Studies

To evaluate the robustness of the central claims, we conduct a series of controlled ablation studies while keeping all other parameters fixed.

**Timestep Selection**   We vary the optimized diffusion timestep to study the trade-off between spatial accuracy and image fidelity across early, intermediate, and late diffusion stages.

**LoRA Fine-Tuning Strength**   We analyze the effect of identity-preserving fine-tuning by varying the number of LoRA training steps and comparing against a no-LoRA baseline.

**Mask Regularization Weight**   We vary the mask regularization coefficient $\lambda$ to examine its influence on spatial precision and visual consistency.

**UNet Feature Supervision**   Motion supervision is applied to different UNet decoder blocks to assess the impact of feature abstraction level on spatial accuracy.

**Multi-Timestep Latent Optimization**   We additionally implement a multi-timestep variant that jointly optimizes multiple diffusion latents within a single editing run, allowing us to directly test the necessity of the single-timestep optimization strategy proposed by the authors.

### 3.5 Evaluation Metrics

We follow the DragBench evaluation protocol used in (Shi et al., 2024) and report two quantitative metrics.

**Mean Distance (MD)**   Mean Distance measures the average Euclidean distance between the target points and their final semantic locations after editing. This metric follows the definition introduced in DragGAN (Pan et al., 2023) and evaluates spatial accuracy of the drag operation.

**Image Fidelity (IF)**   Image Fidelity is defined as $1 - \text{LPIPS}$, where LPIPS is a learned perceptual similarity metric introduced by (Zhang et al., 2018). Higher IF values indicate better preservation of the original image appearance.

### 3.6 Computational Setup

**Hardware:**

- **GPU:** NVIDIA A100 40GB

- **CPU:** AMD EPYC 7453 28-Core Processor

- **RAM:** 64 GB

- **Operating System:** Ubuntu 22.04.3 LTS (Jammy Jellyfish)

**Software Environment:**

- **Python:** 3.8.5

- **CUDA:** 11.7 (via `cudatoolkit` 11.7.0)

- **PyTorch:** 2.0.0

- **diffusers:** 0.24.0

- **transformers:** 4.27.0

**Runtime Statistics:**

- **Average time per sample:** approximately 7 seconds

- **LoRA training time:** approximately 1.8 minutes per sample (80 fine-tuning steps)

- **Total evaluation time:** approximately 7.5 hours

- **Estimated compute budget:** approximately 9 GPU-hours

### 3.7 Ambiguities and Implementation Choices

Although DragDiffusion provides a functional reference implementation, several implementation details are not explicitly discussed in the paper and must be fixed to ensure consistent reproduction.

**DDIM Inversion.** The codebase performs deterministic DDIM inversion using a fixed number of steps and scheduler settings defined in the configuration files. The latent used for optimization is extracted at a user-specified timestep from the inverted trajectory. We adhere to the default inversion procedure, including image preprocessing and scheduler parameters, as changes to these settings noticeably affect the quality and stability of downstream latent optimization.

**Random Seed Control.** The repository exposes seed initialization for PyTorch, NumPy, and Python's random module. When these seeds are fixed, we observe consistent qualitative behavior across runs. However, small numerical variations can still arise from GPU execution, particularly during latent optimization and LoRA fine-tuning. All experiments therefore use fixed seeds and identical execution order, and results are compared based on relative trends rather than exact numerical equality.

**Numerical Precision.** By default, the implementation runs most inference and optimization steps in mixed precision to reduce memory usage and improve throughput. We retain the default precision settings used in the released code. Forcing full fp32 execution did not improve spatial accuracy in our tests and increased runtime, while altering precision modes affected convergence behavior during optimization.

**GPU Determinism.** Despite deterministic DDIM inversion and fixed random seeds, full numerical determinism is not guaranteed due to non-deterministic CUDA and cuDNN kernels. We do not enable strict deterministic backend flags, as this deviates from the intended execution environment and substantially degrades performance. Instead, we focus on reproducing stable qualitative behavior and consistent quantitative trends across configurations.

These implementation choices reflect a faithful execution of the released codebase and clarify which aspects of DragDiffusion are sensitive to low-level configuration during reproduction.

# 4 Experiments and Results

## 4.1 Claim 1: Effect of Diffusion Timestep Selection (Original: Fig. 7(a), (Shi et al., 2024))

A key design choice in DragDiffusion is to perform latent optimization at a single diffusion timestep rather than along the full diffusion trajectory. The authors argue that selecting an intermediate timestep provides sufficient semantic and geometric structure for accurate point-based manipulation, while avoiding the rigidity of early timesteps and the instability of highly noisy latents (Shi et al., 2024). We reproduce this claim by evaluating DragDiffusion under different optimized timesteps while keeping all other settings fixed.

Specifically, we vary the optimized timestep $t \in \{20, 35, 50\}$, corresponding to early, intermediate, and late stages of the diffusion process. For all settings, the number of DDIM steps is fixed to 50, identity-preserving LoRA fine-tuning is enabled, and the same images, prompts, drag instructions, and random seed are used.

Table 1 reports the quantitative results. Optimizing at the intermediate timestep ($t = 35$) achieves the lowest Mean Distance, indicating the most accurate alignment between handle points and their intended target locations. Earlier optimization ($t = 20$) results in higher Image Fidelity, suggesting stronger preservation of the original appearance, but leads to worse spatial accuracy due to limited latent flexibility. In contrast, optimizing at a later timestep ($t = 50$) substantially degrades both Mean Distance and Image Fidelity, reflecting the difficulty of performing stable optimization in highly noisy latent representations. Overall, these results closely match the trends reported in the original work and confirm that a single, well-chosen intermediate timestep provides the best trade-off between spatial controllability and image fidelity in DragDiffusion. This behavior is consistent with the original DragDiffusion results (Fig. 7(a), (Shi et al., 2024)), which identify an intermediate diffusion timestep (t ≈ 30–40) as optimal for balancing spatial accuracy and image fidelity.

Table 1: Effect of optimized diffusion timestep on DragDiffusion performance. Lower Mean Distance (MD) and higher Image Fidelity (IF) indicate better performance. This experiment reproduces the timestep ablation reported in Fig. 7(a) of (Shi et al., 2024).

| Timestep $t$ | Mean Distance (MD) ↓ | Image Fidelity (IF) ↑ |
|---|---|---|
| 20 | 40.90 | 0.8911 |
| **35** | **34.90** | 0.8466 |
| 50 | 49.30 | 0.7666 |

## 4.2 Claim 2: Effect of Identity-Preserving LoRA Fine-Tuning (Original: Fig. 8, (Shi et al., 2024))

DragDiffusion relies on identity-preserving fine-tuning using Low-Rank Adaptation (LoRA) to stabilize drag-based edits and preserve object appearance (Shi et al., 2024; Hu et al., 2022). To evaluate the necessity of this component, we compare DragDiffusion with LoRA enabled against an otherwise identical setting without LoRA.

All experiments use DDIM steps fixed to 50 and an optimized timestep of $t = 35$. Both settings are evaluated on the same DragBench subset using identical images, prompts, drag instructions, and random seed.

Table 2 reports the quantitative results. Enabling LoRA substantially improves spatial accuracy, reducing Mean Distance from 55.68 to 34.90 (a relative improvement of approximately 37%). Image fidelity also improves, with Image Fidelity increasing from 0.8646 to 0.8822. Without LoRA, the model exhibits significantly worse point alignment and reduced identity preservation, indicating that fine-tuning is essential for stable drag-based editing. These findings are consistent with the original DragDiffusion results and confirm that identity-preserving fine-tuning is a critical component of the method.

Table 2: Comparison of DragDiffusion with and without identity-preserving LoRA fine-tuning. Lower Mean Distance (MD) and higher Image Fidelity (IF) indicate better performance. This corresponds to the component ablation shown in Fig. 8 of (Shi et al., 2024).

| Setting | Mean Distance (MD) ↓ | Image Fidelity (IF) ↑ |
|---|---|---|
| Without LoRA | 55.68 | 0.8646 |
| With LoRA | **34.90** | **0.8822** |

### 4.3 Claim 3: Effect of LoRA Fine-Tuning Steps (Original: Fig. 7(b), (Shi et al., 2024))

DragDiffusion employs identity-preserving fine-tuning using Low-Rank Adaptation (LoRA) to stabilize drag-based edits and preserve object identity (Shi et al., 2024; Hu et al., 2022). The original implementation uses 80 LoRA fine-tuning steps as a default setting. To analyze the sensitivity of performance to the extent of fine-tuning, we reproduce the LoRA-step ablation by varying the number of training steps while keeping all other parameters fixed (DDIM=50, optimized timestep $t = 35$).

Table 3 reports the quantitative results for LoRA steps $\{0, 20, 40, 80, 100, 120\}$. Introducing LoRA leads to a substantial improvement in spatial accuracy, with Mean Distance decreasing from 54.60 (no LoRA) to 41.04 at 20 steps. Performance continues to improve with additional fine-tuning and reaches its best value at 100 steps (MD 34.55), slightly outperforming the default 80-step setting (MD 34.90). Beyond this point, further fine-tuning yields diminishing returns and degrades performance at 120 steps. Image Fidelity follows a similar trend, improving significantly once LoRA is enabled and peaking at 100 steps. As in the original DragDiffusion results (Fig. 7(b), (Shi et al., 2024)), performance improves rapidly with early LoRA fine-tuning and plateaus beyond 80–100 steps, indicating diminishing returns from prolonged fine-tuning.

Table 3: Effect of the number of LoRA fine-tuning steps (DDIM=50, $t = 35$). Lower Mean Distance (MD) and higher Image Fidelity (IF) indicate better performance. This reproduces the LoRA-step ablation shown in Fig. 7(b) of (Shi et al., 2024).

| LoRA Steps | Mean Distance (MD) ↓ | Image Fidelity (IF) ↑ |
|---|---|---|
| 0 | 54.6062 | 0.8340 |
| 20 | 41.0366 | 0.8534 |
| 40 | 35.9723 | 0.8497 |
| 80 | 34.8997 | 0.8466 |
| **100** | **34.5541** | **0.8820** |
| 120 | 35.5443 | 0.8756 |

To qualitatively illustrate the effect of LoRA fine-tuning strength, we visualize the editing results for a representative DragBench example under different LoRA step settings (Figure 3).

### 4.4 Claim 4: Effect of Mask Regularization Strength (Original: Eq.(3) and Sec.4.1, (Shi et al., 2024))

DragDiffusion incorporates a spatial mask regularization term to restrict latent updates to user-specified editable regions and prevent unintended global distortions (Shi et al., 2024). The importance of constraining edits to localized regions has also been emphasized in prior interactive image editing methods, where spatial or attention-based masking is used to stabilize edits and preserve overall image structure (Pan et al., 2023; Hertz et al., 2023). In the original DragDiffusion implementation, the mask regularization weight is set to $\lambda = 0.1$.

To evaluate the impact of mask regularization strength and validate the original design choice, we vary $\lambda \in \{0.0, 0.1, 0.5, 1.0\}$ while keeping all other parameters fixed. All experiments use DDIM steps fixed to

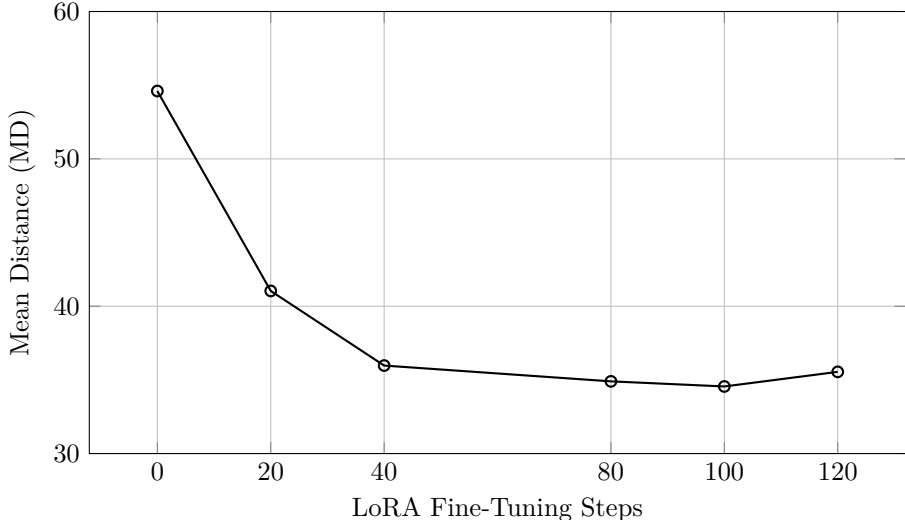

Figure 2: Mean Distance as a function of the number of LoRA fine-tuning steps. Performance improves rapidly with early fine-tuning and peaks at 100 steps, while gains beyond 80 steps are modest. The qualitative trend matches Fig. 7(b) in (Shi et al., 2024).

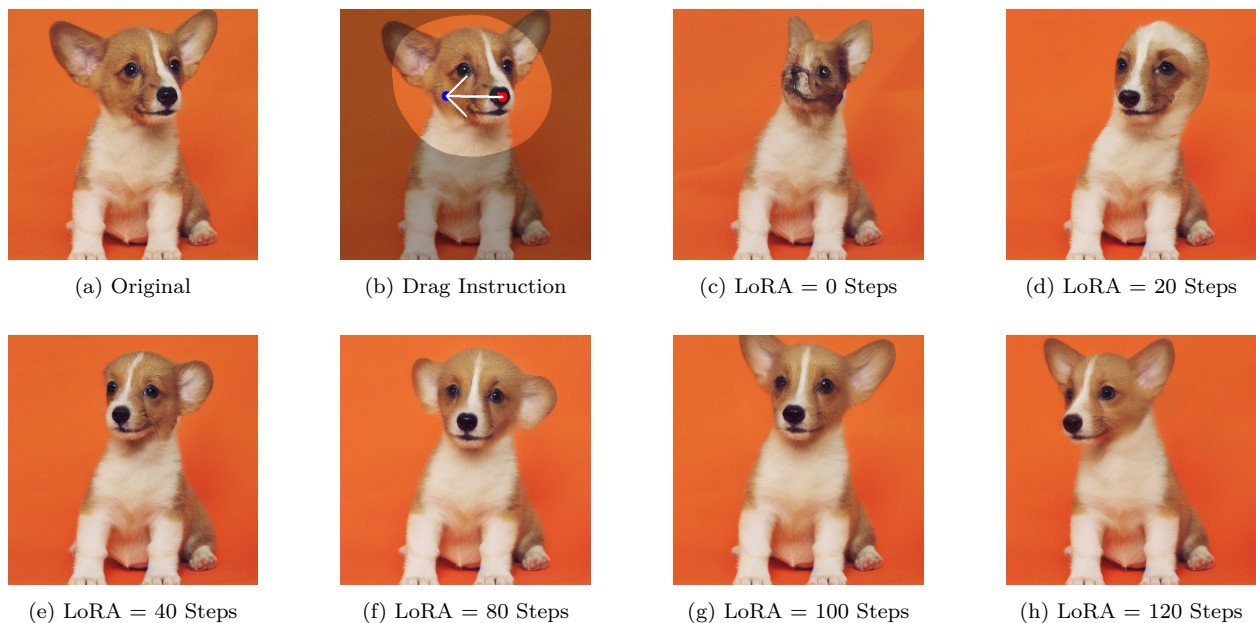

Figure 3: Qualitative effect of the number of LoRA fine-tuning steps on drag-based editing. All results correspond to the same input image and drag instruction. Increasing the number of LoRA steps improves edit stability and identity preservation, with minimal visible differences beyond 80–100 steps.

50, an optimized timestep of $t = 35$, and identity-preserving LoRA fine-tuning enabled. Evaluations are performed on the same DragBench subset using identical images, prompts, drag instructions, and random seed.

Table 4 reports the quantitative results. Without mask regularization ($\lambda = 0$), performance degrades substantially, with higher Mean Distance and reduced Image Fidelity, indicating uncontrolled edits and background distortions. Introducing moderate regularization ($\lambda = 0.1$), which matches the original implementation, yields the lowest Mean Distance and achieves the most accurate point alignment. Stronger regularization

Table 4: Effect of mask regularization strength $\lambda$ (DDIM=50, $t = 35$, LoRA enabled). Lower Mean Distance (MD) and higher Image Fidelity (IF) indicate better performance. This experiment corresponds to the mask regularization analysis discussed in Sec. 4.1 of (Shi et al., 2024).

| $\lambda$ | Mean Distance (MD) $\downarrow$ | Image Fidelity (IF) $\uparrow$ |
|---|---|---|
| 0.0 | 36.9804 | 0.7640 |
| **0.1** | **34.1272** | 0.8736 |
| 0.5 | 34.8613 | 0.8771 |
| 1.0 | 36.6451 | **0.8823** |

($\lambda \geq 0.5$) further improves Image Fidelity but slightly worsens spatial accuracy, reflecting a trade-off between preserving global appearance and allowing flexible local motion. Overall, these results closely match the trends reported in (Shi et al., 2024) and confirm that moderate mask regularization is essential for stable and precise drag-based editing.

## 4.5 Claim 5: Effect of UNet Feature Supervision Layer (Original: Fig. 7(c), (Shi et al., 2024))

DragDiffusion applies motion supervision by comparing UNet feature representations at handle and target point locations during latent optimization. A key design choice is which UNet decoder block is used for this supervision. Prior work on convolutional neural networks has shown that feature representations are hierarchical: shallower layers encode coarse, high-level semantic structure, while deeper layers capture fine-grained texture and local appearance (Zeiler & Fergus, 2014; Bau et al., 2017). Motivated by this hierarchy, DragDiffusion hypothesizes that mid-level UNet features provide the best balance between semantic guidance and spatial precision for point-based manipulation (Shi et al., 2024).

To reproduce and analyze this design choice, we evaluate DragDiffusion using feature maps from decoder blocks $\{1, 2, 3, 4\}$ for motion supervision, while keeping all other parameters fixed (DDIM=50, optimized timestep $t = 35$, mask regularization $\lambda = 0.1$, and identity-preserving LoRA fine-tuning enabled). Lower-index decoder blocks correspond to more semantic but spatially coarse representations, whereas higher-index blocks emphasize local texture and appearance. This experiment directly corresponds to Fig. 7(c) in the original DragDiffusion paper, which evaluates spatial accuracy and image fidelity across UNet decoder blocks.

Table 5 reports the quantitative results. Supervision at the third decoder block yields the lowest Mean Distance, indicating the most accurate alignment between handle points and their intended target locations. In contrast, supervision at the first decoder block performs poorly, suggesting that overly coarse semantic features lack sufficient spatial resolution for precise point-based manipulation. Applying supervision at the deepest decoder block achieves the highest Image Fidelity but significantly worsens Mean Distance, reflecting a bias toward preserving local appearance rather than enforcing global structural motion. Figures 4a and 4b further illustrate this trade-off between spatial accuracy and image fidelity across feature levels.

The original DragDiffusion paper reports the same qualitative behavior: mid-level UNet features yield the best spatial accuracy, while deeper features improve appearance preservation at the cost of precise point alignment (Shi et al., 2024). Our reproduced results exhibit the same ordering across decoder blocks and select the same optimal feature level, confirming the consistency of the reported behavior despite minor differences in evaluation setup and absolute metric values.

## 4.6 Multi-Timestep Latent Optimization (Extension)

A central design principle of DragDiffusion is that accurate and efficient point-based editing can be achieved by optimizing a single diffusion latent at a carefully chosen intermediate timestep. This choice is motivated by the observation that intermediate timesteps balance semantic structure and spatial flexibility, while avoiding excessive noise at early timesteps and over-constrained representations at late timesteps (Shi et al., 2024). To further examine this assumption, we conduct an additional experiment that extends the original method

Table 5: Effect of UNet decoder block used for motion supervision (DDIM=50, $t = 35$, $\lambda = 0.1$, LoRA enabled). Lower Mean Distance (MD) and higher Image Fidelity (IF) indicate better performance. This reproduces the UNet feature ablation reported in Fig. 7(c) of (Shi et al., 2024).

| Decoder Block | Mean Distance (MD) ↓ | Image Fidelity (IF) ↑ |
|:---:|:---:|:---:|
| 1 | 54.4055 | 0.8290 |
| 2 | 36.1107 | 0.8561 |
| **3** | **35.1043** | 0.8734 |
| 4 | 46.1320 | **0.9123** |

(a) Mean Distance          (b) Image Fidelity

Figure 4: Effect of the UNet decoder block used for motion supervision. Mean Distance (left) and Image Fidelity (right) illustrate the trade-off between spatial accuracy and appearance preservation across feature levels. The reproduced ordering of decoder blocks exactly matches the original findings (Fig. 7(c), (Shi et al., 2024), with mid-level features achieving the best spatial accuracy.

by jointly optimizing multiple diffusion latents within a single editing run and does not correspond to a reported result in the original paper.

**Experimental Setup.** Instead of optimizing a single latent at timestep $t = 35$, we optimize a set of latents at timesteps $t_{\text{set}} = \{30, 35, 40\}$. During each optimization step, motion supervision and mask regularization losses are applied independently at each timestep, and gradients are accumulated to update all selected latents. All other settings are kept identical to the baseline configuration, including DDIM steps fixed to 50, mask regularization $\lambda = 0.1$, identity-preserving LoRA fine-tuning enabled, and identical images, prompts, drag instructions, and random seed. Evaluation is performed on the same DragBench subset using the same Mean Distance (MD) and Image Fidelity (IF) metrics.

**Quantitative Results.** Table 6 compares single-timestep and multi-timestep optimization. Multi-timestep optimization achieves comparable Image Fidelity but does not improve spatial accuracy, yielding slightly higher Mean Distance than the single-timestep baseline. In addition, optimizing multiple latents substantially increases computational cost, resulting in significantly longer runtimes per editing operation.

**Discussion.** These results suggest that jointly optimizing multiple diffusion latents does not provide complementary supervision signals beyond what is already captured at an appropriately chosen intermediate timestep. Prior analyses of diffusion dynamics have shown that neighboring timesteps exhibit highly correlated latent representations, particularly in the mid-range of the diffusion process (Song et al., 2021; Karras et al., 2022). As a result, optimizing multiple closely spaced timesteps may introduce redundant constraints rather than additional useful guidance. Moreover, enforcing motion consistency across multiple noise levels

can over-constrain the optimization, limiting the model's ability to flexibly satisfy spatial objectives at any single timestep.

Overall, this experiment reinforces the original design choice of DragDiffusion: single-timestep latent optimization at an intermediate diffusion stage is sufficient to achieve accurate spatial control while maintaining computational efficiency. Extending optimization across multiple timesteps increases complexity and runtime without providing measurable performance benefits.

Table 6: Comparison between single-timestep and multi-timestep latent optimization. Both methods use DDIM=50, $\lambda = 0.1$, $t = 35$ (baseline), and LoRA enabled.

| Method | Mean Distance (MD) $\downarrow$ | Image Fidelity (IF) $\uparrow$ | Runtime (s) $\downarrow$ |
|---|---|---|---|
| Single timestep ($t = 35$) | **35.10** | 0.8734 | **1.0$\times$** |
| Multi-timestep ($\{30, 35, 40\}$) | 36.28 | 0.8719 | 2.7$\times$ |

## 5 Conclusion

This paper presented a comprehensive reproducibility study of DragDiffusion, a diffusion-based framework for interactive point-based image editing. Using the authors' released implementation and the DragBench benchmark, we systematically reproduced the key experimental claims of the original work and evaluated the robustness of its central design choices under controlled settings. Our goal was not to improve performance, but to verify the validity of the proposed methodology and assess the sensitivity of its core components.

Across all five primary claims, our reproduced results closely match the qualitative and quantitative trends reported in the original DragDiffusion paper. We confirm that optimizing a single diffusion latent at an appropriately chosen intermediate timestep is sufficient for accurate spatial control, and that identity-preserving LoRA fine-tuning is essential for maintaining object identity during drag-based edits. We further verify that moderate levels of LoRA fine-tuning and mask regularization provide strong performance, with diminishing returns observed beyond recommended defaults. In addition, our experiments confirm that motion supervision applied to mid-level UNet decoder features yields the best balance between spatial accuracy and image fidelity, aligning with both the original implementation and established feature hierarchy principles.

Beyond reproducing the original claims, our study highlights several practical sensitivities relevant for future work. We find that timestep selection and feature-level supervision are particularly influential, with small deviations leading to noticeable performance degradation. In contrast, other hyperparameters, such as LoRA training duration and mask regularization strength, exhibit broader operating ranges once reasonable values are chosen. These observations suggest that while DragDiffusion is robust when configured correctly, careful attention must be paid to a small number of critical design decisions to achieve stable and accurate results.

We also explored a multi-timestep latent optimization variant as an extension beyond the original methodology. Our results show that jointly optimizing multiple diffusion latents does not improve spatial accuracy or image fidelity compared to the single-timestep baseline, while significantly increasing computational cost. This finding reinforces the original design choice of DragDiffusion and suggests that the benefits of multi-timestep supervision are limited due to redundancy across neighboring diffusion states.

Finally, we acknowledge several limitations of our reproduction. Our evaluation is restricted to the DragBench dataset and to Stable Diffusion v1.5, and absolute metric values may vary under different hardware configurations or random seeds. Nevertheless, the consistency of observed trends across all experiments provides strong evidence that the central claims of DragDiffusion are robust and reproducible. Overall, this study validates DragDiffusion as an effective and well-motivated approach to interactive image editing, while providing practical insights into the factors that most strongly influence its performance.

### 5.1 What was easy to reproduce.

From a reproducibility standpoint, several aspects of DragDiffusion were straightforward to replicate. The authors provide a complete and well-structured codebase, along with the DragBench benchmark and clearly defined evaluation metrics. Core components of the pipeline, including DDIM inversion, single-timestep latent optimization, and motion supervision, could be reproduced with minimal modification. In addition, the reported qualitative and quantitative trends for key design choices—such as timestep selection, mask regularization, and UNet feature supervision—were consistently observed in our experiments, indicating that the method is stable when configured according to the original implementation.

### 5.2 What was difficult or sensitive.

At the same time, we encountered several challenges that are important to highlight. While the overall pipeline is reproducible, performance is sensitive to a small number of hyperparameters, including the optimized diffusion timestep, the strength of mask regularization, and the feature level used for motion supervision. Minor deviations from the recommended settings can lead to noticeable degradation in spatial accuracy or image fidelity. In addition, identity-preserving LoRA fine-tuning introduces non-trivial computational overhead and requires careful checkpoint management to ensure consistent behavior across experiments. We also observed that the released codebase depends on several library versions that have since evolved, requiring manual configuration and adaptation to newer versions of core dependencies to ensure compatibility and correct execution. These factors do not prevent reproduction, but they underscore the importance of careful experimental control, environment configuration, and detailed reporting when applying DragDiffusion in practice.

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

# A   Additional Qualitative Results

## A.1   Mask Regularization

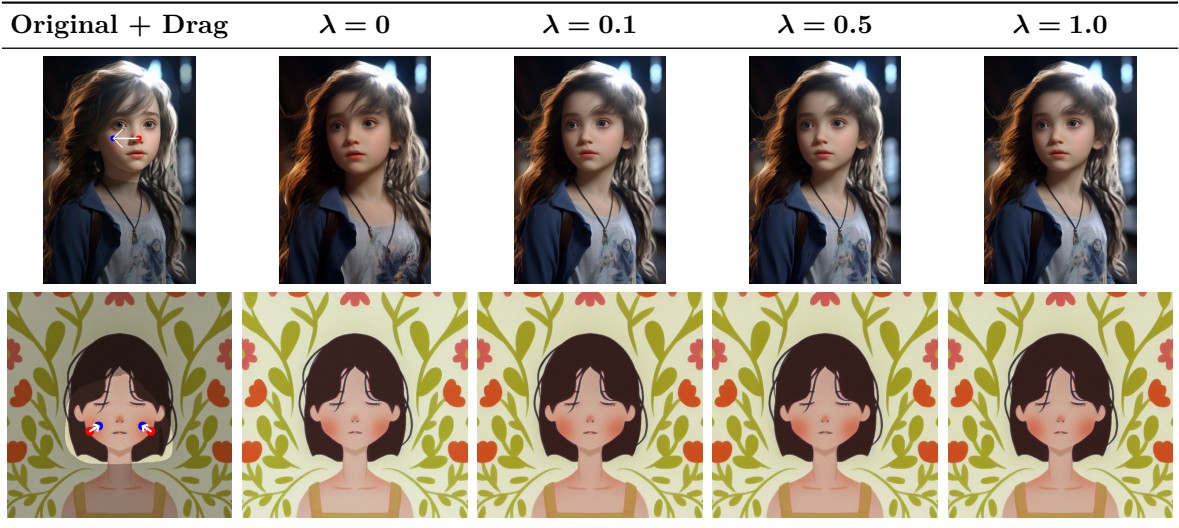

Figure 5: Qualitative comparison of mask regularization strength $\lambda$. Without regularization ($\lambda = 0$), background distortions are visible due to unconstrained latent updates. Moderate regularization ($\lambda = 0.1$) provides the best balance between accurate point manipulation and preservation of non-edited regions. Stronger regularization ($\lambda \geq 0.5$) increasingly restricts motion, leading to ineffective edits despite improved image fidelity.

## A.2   UNet decoder block qualitative comparison

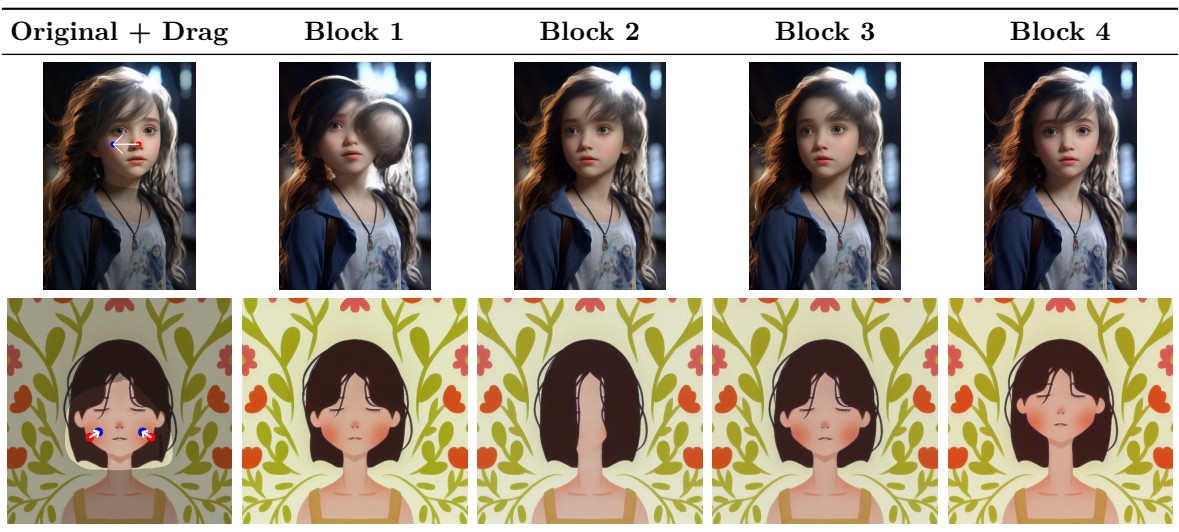

Figure 6: Qualitative comparison of motion supervision applied to different UNet decoder blocks. Supervision at shallow features (Block 1) often fails to produce effective point-based motion, while very deep features (Block 4) tend to preserve appearance but can underperform in enforcing the desired geometric change. Consistent with the quantitative results, mid-level features (Block 3) provide the most reliable balance between spatial control and visual fidelity.

## A.3 Effect of LoRA Fine-Tuning Steps

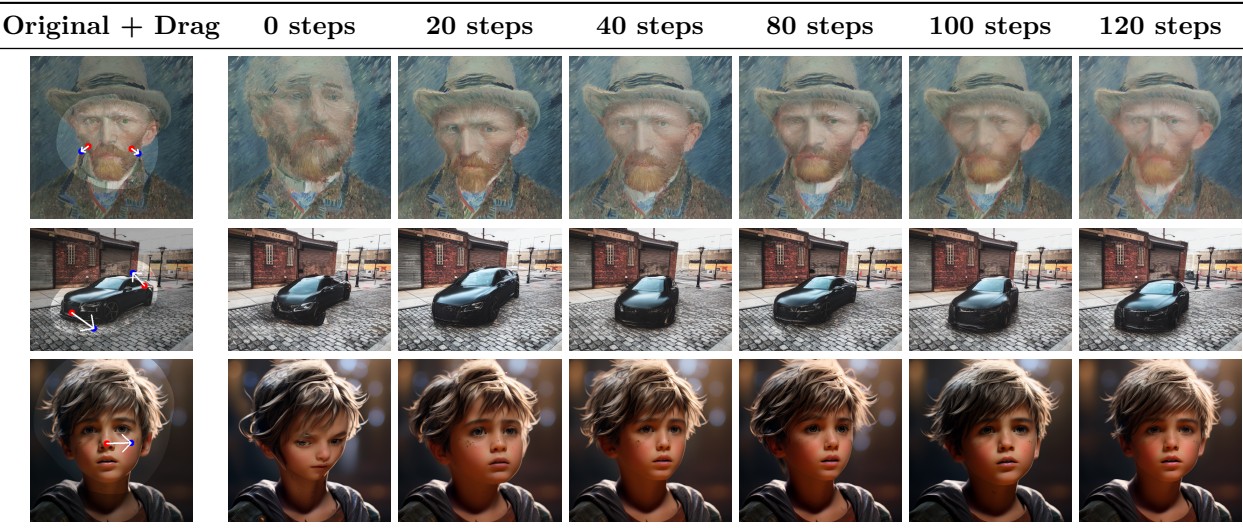

Figure 7: Qualitative comparison illustrating the effect of increasing the number of LoRA fine-tuning steps. Without LoRA (0 steps), edits often exhibit identity drift and unstable local appearance. Increasing the number of fine-tuning steps progressively improves identity preservation and edit stability, with diminishing visual gains beyond 80–100 steps.

## A.4 Effect of Multi-Timestep Latent Optimization

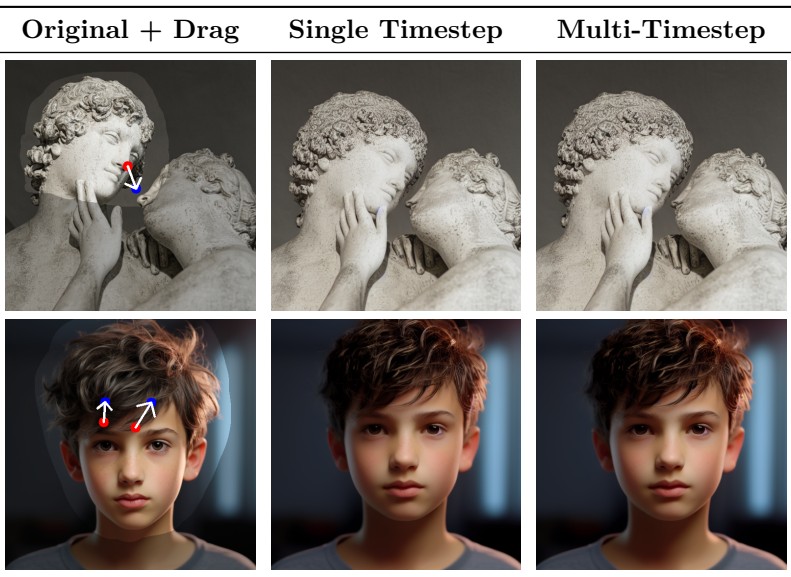

Figure 8: Qualitative comparison between single-timestep and multi-timestep latent optimization. While multi-timestep optimization produces visually similar results to the single-timestep baseline, it does not lead to improved spatial alignment. This observation is consistent with the quantitative results, which show increased Mean Distance and substantially higher runtime for the multi-timestep variant.

# B  Dataset

All experiments in this reproducibility study are conducted on *DragBench*, the benchmark dataset released as part of the DragDiffusion codebase. DragBench is specifically designed to evaluate interactive point-based image editing methods under controlled and reproducible settings. Each sample consists of an input image, a text prompt describing the image content, a binary edit mask defining the editable region, and a set of handle–target point pairs specifying the desired spatial manipulation.

The dataset spans a diverse range of image categories, including humans, animals, buildings, landscapes, and objects, enabling evaluation across varying semantic complexity and structural characteristics. For each sample, the drag instruction is predefined and remains fixed across all experimental conditions, ensuring fair and consistent comparison between different model variants and ablation settings.

In our experiments, we use the DragBench subset provided with the official DragDiffusion implementation without modification. All samples are processed using identical preprocessing, prompts, drag instructions, and random seeds unless explicitly stated otherwise. This ensures that observed performance differences arise solely from changes in the evaluated method components rather than dataset variations.

