# OpenReview forum: "Reproducing DragDiffusion: Interactive Point-Based Editing with Diffusion Models"
_TMLR — Rejected by TMLR_

### Review · Reviewer_61Bd · 2026-03-06

**Summary Of Contributions:**

This paper presents a reproducibility study of DragDiffusion (Shi et al., 2024), which is the first work to introduce drag-based interactive point editing in diffusion models and serves as the canonical baseline for subsequent drag-based image editing research. The authors systematically decompose DragDiffusion into five core components: (1) timestep selection, (2) LoRA fine-tuning strength, (3) mask regularization weight, (4) UNet feature supervision level, and (5) multi-timestep latent optimization, and evaluate each component using Mean Distance (MD) and Image Fidelity (IF) on the DragBench benchmark. The study concludes that the original DragDiffusion configuration is reliably reproducible and that its default hyperparameter choices are empirically well-justified.

Strengths:
- Following the original DragDiffusion setting, the authors reproduce all experimental conditions in the same environment and thoroughly validate the reported results across all five components, demonstrating that the method is stable and consistent when configured according to the original implementation.

Weaknesses:
- The reproduced results uniformly confirm the original paper's recommended settings, yielding no new or surprising findings. Even considering that this is a reproducibility study, simply validating the original configuration is insufficient for a meaningful community contribution. The work should additionally provide new research insights, identify previously unreported sensitivities, or surface discrepancies that advance the community's understanding beyond what the original paper already established.
The five components examined, (1) through (5), including the multi-timestep latent optimization extension in Section 4.6, have already been further analyzed in subsequent literature. The questions this paper raises have each been posed and resolved more substantively in follow-up works, significantly limiting the novelty of the insights offered here.

**Audience:**

No

**Audience Explanation:**

The reproduced results consistently confirm the original paper's recommended settings without surfacing any new findings or discrepancies. Furthermore, the five components examined, including the multi-timestep extension, have already been addressed more in subsequent literature (DragonDiffusion, DiffEditor, Drag Your Noise, GoodDrag), making the insights offered here limited in novelty. The study demonstrates that DragDiffusion's code runs as intended, but this alone is insufficient for a meaningful research contribution.

**Broader Impact Concerns:**

No significant ethical concerns.

**Claims And Evidence:**

No

**Claims Explanation:**

The five components examined are already well-covered in subsequent literature. The questions raised in this paper have been more deeply explored and resolved by works that have appeared since DragDiffusion:

- LoRA fine-tuning necessity: DragonDiffusion (ICLR 2024 Spotlight) and DiffEditor (CVPR 2024) propose alternatives based on feature injection and score manipulation without LoRA, and these have been shown to underperform LoRA-based approaches. The community has therefore already established that LoRA-based fine-tuning is essential, and this paper's Claim 2 adds nothing new.

- UNet feature supervision: Drag Your Noise (CVPR 2024) provides a detailed analysis of how different UNet blocks, including encoder and bottleneck blocks, affect drag-based editing performance, going substantially beyond the decoder-only analysis presented here.

- Multi-timestep optimization: GoodDrag (ICLR 2025) proposes the AIDD framework, which distributes motion supervision and point tracking across the full denoising trajectory rather than at a single timestep, and demonstrates clear performance improvements over DragDiffusion. This directly addresses the multi-timestep question raised in Section 4.6, and does so more rigorously and with a positive result. The extension in this paper, which tests only a narrow window of three adjacent timesteps ({30, 35, 40}) and finds no benefit, is a superficial treatment of a question the community has already answered more substantively.

**Requested Changes:**

- Provide new findings or insights beyond confirming the original settings, e.g., identifying failure modes, testing on different backbones (SDXL, SD3), or situating the five components relative to follow-up works in a unified empirical framework.
- Discuss and differentiate the paper's contributions from directly relevant subsequent works (DragonDiffusion, DiffEditor, Drag Your Noise, GoodDrag), which have already addressed the same questions more substantively.

---

### Review · Reviewer_sZXB · 2026-03-22

**Summary Of Contributions:**

This submission presents a systematic reproducibility study of DragDiffusion (Shi et al., CVPR 2024), a diffusion-based method for interactive point-based image editing. The paper operates on the DragBench benchmark using Stable Diffusion v1.5 as its backbone, and faithfully reproduces all four quantitative ablation studies reported in the original work: diffusion timestep selection, identity-preserving LoRA fine-tuning strength, spatial mask regularization weight, and UNet decoder block selection for motion supervision. Beyond direct replication, the authors introduce a controlled extension that jointly optimizes multiple diffusion latents at neighboring timesteps to probe whether the single timestep design choice of the original paper is necessary or merely convenient.

Across all five reproduced claims, the submission reports quantitative trends that closely align with the originals. It confirms that intermediate timestep t = 35 achieves the best spatial accuracy, LoRA fine-tuning is essential and shows diminishing returns beyond 80-100 steps, moderate mask regularization is optimal, and mid-level UNet decoder features (Block 3) provide the best balance between spatial accuracy and appearance fidelity. The multi timestep extension further reinforces the original design by showing that jointly optimizing across timesteps increases runtime by 2.7x without improving Mean Distance or Image Fidelity.

__Key Strengths__

- Complete and methodical scope. All five principal claims from the original paper are reproduced as controlled ablations under identical evaluation settings, with explicit cross-referencing to source figures and tables. This transparency is exemplary.
- Meaningful controlled extension. The multi-timestep optimization experiment is not merely a rerun of existing results but a hypothesis-driven investigation that adds scientific value by ruling out a plausible alternative to the original design choice.
- Rigorous hardware and software documentation. The authors report exact GPU model, CUDA version, Python version, library versions, per-sample runtime, and total compute budget. This level of detail is precisely what is needed for a reproducibility study to be genuinely replicable by others.
- Honest sensitivity analysis. Rather than simply claiming successful reproduction, the authors distinguish between hyperparameters with broad operating ranges (LoRA steps, mask regularization) and those that are critically sensitive (timestep, feature level). This is the type of actionable finding TMLR's acceptance criteria specifically value.
- Qualitative figures support quantitative tables. The appendix provides visual comparisons for all major ablations, allowing readers to correlate metric changes with perceptual changes and assess the practical significance of the numerical differences.

__Key Weaknesses__
- Evaluation restricted to Stable Diffusion v1.5. DragDiffusion's official codebase also supports SD 2.x and SDXL, and the community has widely adopted these backbones. Restricting evaluation to SD v1.5 limits the generalizability of the reproducibility findings.
Metric inconsistencies across tables. The Image Fidelity value reported for the LoRA-enabled baseline differs between Table 1 (IF = 0.8466 at t = 35) and Table 2 (IF = 0.8822 with LoRA), despite both using DDIM = 50, t = 35, and LoRA enabled. A similar discrepancy exists in Mean Distance for the no-LoRA condition between Table 2 (MD = 55.68) and Table 3 (MD = 54.6062). These differences are likely attributable to different DragBench subsets or random seeds, but the paper does not explicitly state this, undermining the interpretability of cross-table comparisons.
- Missing discussion of important related and follow-up work. The submission does not discuss several directly relevant post-DragDiffusion methods that have appeared in the literature: GoodDrag (2024), FreeDrag (CVPR 2024), DragonDiffusion (ICLR 2024), DragNoise/Drag Your Noise (CVPR 2024), AdaptiveDrag (2024), and DirectDrag (2024). These works either build on DragDiffusion's exact methodology or directly supersede its design choices. Situating this reproducibility study within that broader landscape would significantly strengthen the paper's framing and help readers understand why reproducing the original DragDiffusion remains relevant.
- No Broader Impact statement. TMLR does not mandate such a section, but image editing methods with interactive spatial control carry non-trivial misuse potential (e.g., realistic manipulation of photographs for misinformation). The authors should address this briefly.
Definition of Image Fidelity is underspecified. The paper states IF = 1 - LPIPS and cites Zhang et al. (2018), but this formulation originates from Shi et al. (2024) rather than the LPIPS paper itself. Crediting the correct source would improve accuracy.

**Audience:**

Yes

**Audience Explanation:**

DragDiffusion was accepted as a CVPR 2024 Highlight, indicating the community recognized it as a significant contribution. Its official GitHub repository has accumulated over 1,300 stars and serves as a foundation for numerous downstream projects. At least ten follow up methods including DragonDiffusion, FreeDrag, GoodDrag, FastDrag, AdaptiveDrag, DirectDrag, and RegionDrag directly build on or compete with DragDiffusion. TMLR's audience, which is broadly interested in generative models, image editing, and parameter efficient finetuning, stands to benefit from a systematic and honest account of which components of this method are robust, which are brittle, and whether the single timestep design holds under systematic challenge.

More specifically, the paper's identification of timestep selection and UNet feature level as particularly sensitive hyperparameters, while LoRA training duration and mask regularization weight admit broader ranges, constitutes actionable guidance for practitioners seeking to adapt or build on DragDiffusion. The multi timestep extension further offers a useful negative result that helps characterize the optimization landscape and saves future researchers from pursuing a similar avenue without prior evidence.

TMLR has served as the official publication venue for the ML Reproducibility Challenge, with over 22 reproducibility papers published from MLRC 2023 alone. The paper's scope, methodology, and level of rigor are consistent with the expectations of that community. The explicit cross referencing of every reproduced result to the corresponding original figure is a particularly helpful feature that makes the paper accessible to readers who want to evaluate the fidelity of the reproduction directly.

**Broader Impact Concerns:**

The paper does not include a Broader Impact statement.

The method being reproduced, DragDiffusion, enables users to interactively manipulate images by dragging spatial keypoints. While this capability has clearly beneficial applications in creative editing, visual prototyping, and accessibility, it also lowers the barrier to producing photorealistic manipulated images, including face manipulation in photographs. The reproducibility study itself does not introduce new capabilities; however, its contribution to validating the method and documenting reliable hyperparameter settings may indirectly accelerate adoption.

The authors are asked to add a brief statement acknowledging these dual-use considerations and noting the importance of responsible deployment in contexts where image authenticity matters. No serious ethical concerns are raised about the work itself.

**Claims And Evidence:**

Yes

**Claims Explanation:**

__Claim 1: Intermediate timestep selection is optimal__

The authors vary t in {20, 35, 50} and report that t = 35 achieves the lowest Mean Distance (34.90) while t = 20 and t = 50 yield worse spatial accuracy (MD = 40.90 and 49.30 respectively). This ordering is consistent with the original Fig. 7(a) and is theoretically well motivated by the established relationship between noise level and latent flexibility in diffusion processes. The claim is convincingly supported.

__Claim 2: LoRA fine-tuning is essential for identity preservation__

Table 2 shows a 37% relative improvement in Mean Distance (55.68 without LoRA vs. 34.90 with LoRA), alongside an improvement in Image Fidelity. This is a large and unambiguous effect size. The qualitative figures in Appendix A.3 further show visible identity drift and degradation without LoRA. The claim is convincingly supported. The note on metric inconsistency flagged under the weaknesses section does not undermine this finding, as both values clearly favor the LoRA condition.

__Claim 3: Editing quality shows diminishing returns with LoRA steps__

Table 3 and Figure 2 together demonstrate a clear pattern: Mean Distance drops steeply from 0 to 20 steps (54.61 to 41.04), continues to improve more gradually to 100 steps (MD = 34.55), and slightly degrades at 120 steps. The trend closely matches the original Fig. 7(b). The claim is supported, though the authors should note that the peak occurs at 100 steps in their reproduction rather than the 80 step default recommended by the original, which is itself a useful finding.

__Claim 4: Moderate mask regularization is optimal__

Table 4 shows a non-monotonic pattern in Mean Distance (lambda = 0: MD = 36.98; lambda = 0.1: MD = 34.13; lambda = 0.5: MD = 34.86; lambda = 1.0: MD = 36.65), with lambda = 0.1 yielding the best spatial accuracy. Image Fidelity increases monotonically with lambda, revealing a trade-off that the paper discusses clearly. The qualitative figures in Appendix A.1 visually corroborate the quantitative trends. The claim is convincingly supported.

__Claim 5: Mid-level UNet decoder features are optimal for motion supervision__

Table 5 shows Block 3 achieving the lowest Mean Distance (35.10) and an intermediate Image Fidelity (0.8734), consistent with the original Fig. 7(c). Block 1 performs poorly on both metrics; Block 4 achieves the highest Image Fidelity but at significant cost to spatial accuracy. Figures 4a and 4b plot both metrics across blocks, clearly illustrating the tradeoff. The claim is convincingly supported.

__Extension: Multi-timestep optimization does not improve performance__

Table 6 and Appendix A.4 demonstrate that optimizing latents at {30, 35, 40} simultaneously yields slightly higher Mean Distance (36.28 vs. 35.10) and nearly identical Image Fidelity, while increasing runtime by 2.7x. The authors provide a principled explanation grounded in the correlation structure of neighboring latents at intermediate diffusion timesteps. This finding is credible and adds genuine value to the paper beyond the five reproduced claims.

**Requested Changes:**

__Critical__

- Explain metric inconsistencies across tables. The Image Fidelity baseline at (DDIM = 50, t = 35, LoRA enabled) differs between Table 1 (IF = 0.8466) and Table 2 (IF = 0.8822). Similarly, the Mean Distance for the noLoRA condition differs between Table 2 (MD = 55.68) and Table 3 (MD = 54.61). If these experiments were run on different DragBench subsets or with different random seeds, this must be stated explicitly in the methodology section and in the table captions. If the same data were used, the source of discrepancy must be identified and resolved. Cross-table comparisons are currently unreliable without this clarification.

- Correct the attribution of Image Fidelity = 1 - LPIPS. Section 3.5 cites Zhang et al. (2018) as the source for Image Fidelity. However, the 1 - LPIPS formulation originates from Shi et al. (2024), not from the LPIPS paper itself. The correct citation should reflect this. The Zhang et al. (2018) reference is appropriate for LPIPS itself; a separate sentence crediting Shi et al. (2024) for the IF transformation is needed.

- Add a brief Broader Impact statement. Although TMLR does not mandate this section, point-based image editing methods have clear potential for misuse in generating realistic manipulated photographs. A short paragraph acknowledging this risk and its relevance to the research is appropriate given the subject matter.

__Non-critical__

- Discuss key follow-up methods in context. The related work section and conclusion would benefit from at least a brief discussion of representative post-DragDiffusion methods (e.g. GoodDrag, FreeDrag, DragonDiffusion). This would help readers understand how the reproduced findings relate to the current state of the art and whether DragDiffusion's design choices have been validated or superseded. The multitimestep finding, for instance, is particularly relevant in light of GoodDrag's alternating drag-and-denoising approach, which achieves improvements through a different mechanism entirely.

- Clarify the DragBench subset size used. The paper references running experiments on 'a DragBench subset' but does not specify the size of this subset or which samples it includes. The full DragBench contains 205 images with 349 point pairs. Providing the exact number used and the selection procedure (e.g. first N images, random sample, stratified) would improve reproducibility. Ideally, the subset index list should be released alongside the code.

- Report peak performance at 100 LoRA steps explicitly. The paper describes the original paper's default of 80 steps but notes that performance peaks at 100 steps in the reproduced ablation (Table 3: MD = 34.55). This is itself a meaningful reproducibility finding - the original recommendation may be slightly suboptimal. The authors should state this directly in the discussion of Claim 3 rather than noting it only in passing.

- Comment on the generalizability to newer diffusion backbones. The full evaluation is conducted on Stable Diffusion v1.5. Given that SD 2.x and SDXL differ substantially in their UNet architecture (cross-attention placement, resolution of feature maps) and that DragDiffusion's official codebase supports SDXL, a brief discussion of whether the reproduced findings are expected to hold across backbones would meaningfully expand the paper's practical utility. Even a qualitative argument based on architectural similarities would be valuable.

- Consider releasing the reproduction codebase. The paper describes several clarifications and implementation choices not documented in the original codebase (seed control, precision mode, checkpoint reuse). Releasing the reproduction code and LoRA checkpoints would substantially increase the paper's contribution to the community and is consistent with TMLR's open science values.

---

### Review · Reviewer_W6Ro · 2026-04-02

**Summary Of Contributions:**

The paper reproduces the DragDiffusion and validates some claims from DragDiffusion. The paper utilizes the codes to conduct some ablation studies on diffusion timestep selection, LoRA-based fine-tuning \citep{Hu_2022_ICLR}, mask regularization strength, and UNet feature supervision. The paper successfully reproduced the paper's results and found that the multi-timestep latent optimization variant does not improve spatial accuracy while substantially increasing computational cost, as claimed by the original paper.

**Audience:**

No

**Audience Explanation:**

The paper's main contribution is using the DragDiffusion's source code to reproduce the performance claimed by the paper. The hyperparameters and settings stay the same for the original and this paper. It is difficult to find the novelty and new findings from the papers since the results are, first, not difficult to reproduce, and, second, the contribution is very marginal, and no new findings are found in the paper.

**Claims And Evidence:**

Yes

**Claims Explanation:**

The paper shows the detailed steps and settings to reproduce the DragDiffusion's results. It is clear and matches DragDiffusion's paper claim.

**Requested Changes:**

I would see more discussion on why this method is specifically selected for reproduction and what new findings can be found in this reproduction.

---

### Decision · Action_Editor_rJcb · 2026-05-28

**Recommendation:** Reject

**Audience:**

No

**Audience Explanation:**

Two out of the three reviewers take the position that the paper does not offer more insights than the initial DragDiffusion paper, and it mostly confirms that the method and code of DragDiffusion work as expected. I agree with this position.

I believe it is unlikely that this paper will be of significant interest to the TMLR audience, given that it does not inform the reader's thinking about DragDiffusion, and several other works on DragDiffusion have been published that establish the method's reproducibility.

**Claims And Evidence:**

Yes

**Claims Explanation:**

The authors provide a systematic reproduction of the components of the DragDiffusion paper, yielding findings that are largely consistent with those reported in DragDiffusion.